# DiffTune Revisited:
# A Simple Baseline for Evaluating
# Learned llvm-mca Parameters

Andreas Abel

*Saarland University*
*Saarland Informatics Campus*
Saarbrücken, Germany
abel@cs.uni-saarland.de

*Abstract*—Recent work introduced DiffTune, a neural network-based technique to automatically learn the microarchitecture-specific parameters of basic block CPU simulators. The authors apply their approach to the llvm-mca simulator. They show that the learned parameter values achieve an accuracy on the BHive benchmark suite that is comparable, and in some cases even better than that of the original, expert-provided llvm-mca parameter values.

In this paper, we show that an accuracy in this range is actually trivial to achieve: We propose a simple set of parameter values that outperforms the values learned by DiffTune. In fact, our set of parameter values is so simple that it can be fully described within this abstract: We set the dispatch width to 4, the reorder buffer size to 100, the latencies and µop counts of all instructions to 1, and all other parameters to 0. These parameter values lead to more accurate predictions than DiffTune's values on all four microarchitectures that were considered in the DiffTune paper.

We then develop a simple learning algorithm for the llvm-mca parameters. We show that the parameter values learned by our algorithm lead to an average error on the BHive benchmark suite that is between 29% and 47% lower compared to DiffTune's values.

## I. INTRODUCTION

Basic block CPU simulators such as llvm-mca [1], uiCA [4], or CQA [6] are widely used to understand, predict, and optimize the performance of software on x86 systems. Such tools typically support multiple microarchitectures; the details of these microarchitectures are provided via a set of parameters.

Setting the parameter values correctly for a specific microarchitecture can be challenging. The number of parameters is typically high; usually, there are multiple parameters for each instruction variant, such as the latency and the execution port usage, and recent CPUs support several thousand instruction variants. Moreover, the relevant properties are typically not documented. Thus, they are often reverse-engineered using microbenchmarks [2], [3], [8], [9], [13].

Recent work by Renda et al. [12] introduced DiffTune, an alternative approach that aims at overcoming these challenges by learning the microarchitecture-specific parameters of CPU simulators automatically, based only on coarse-grained end-to-end measurements.

DiffTune works as follows. It takes a dataset of benchmarks, in which each benchmark is labeled with performance mea-

surements on the actual hardware. Then, it learns a neural network-based differentiable surrogate for the original simulator, i.e., a differentiable function that takes as input a benchmark from the dataset and a set of parameter values, and outputs an approximation of the output that the original simulator would produce for this input. Finally, it applies gradient-based optimization techniques to find parameter values that minimize the difference between the surrogate's output and the measurements on the benchmarks of the dataset; using such techniques would not be possible on the original, non-differentiable simulator.

Renda et al. apply DiffTune to the llvm-mca simulator, which is a tool that analyzes the performance of basic blocks using information that is available in LLVM. As the dataset, they use the BHive benchmark suite [7], which contains basic blocks that were extracted from applications from different domains, along with throughput measurements for different x86 microarchitectures.

There are, in total, more than $10,000$ llvm-mca parameters that correspond to instructions that occur in the BHive set. Renda et al. show that the values learned by DiffTune for these parameters lead to throughput predictions that have a lower error compared to the measurements than predictions obtained with the original, expert-provided and hand-written llvm-mca parameter values.

In this paper, we will address the question of how difficult it actually is to set the parameters in a way that achieves a comparable accuracy on the BHive benchmark set.

It should be noted that the goal of llvm-mca is not only to predict the throughput but also to provide other insights into how the code is executed. It is therefore possible that if one focuses, as in the DiffTune paper, only on throughput predictions, the problem of setting values that lead to a high accuracy might be easier than the problem the original authors of llvm-mca attempted to solve. Furthermore, a closer inspection of the BHive set reveals that many of the instructions that occur in the benchmarks are relatively simple instructions with a low latency and a small number of µops; more complex instructions occur only in a relatively small number of benchmarks. It is therefore likely that not all of the more than $10,000$ llvm-

TABLE I: llvm-mca parameters

| Parameter | Count | Description |
| --- | --- | --- |
| Dispatch width | 1 global | The maximum number of µops that can be sent from the front end to the scheduler in each cycle. |
| Reorder buffer size | 1 global | The number of entries in the reorder buffer. |
| Latency | 1 per instruction | The number of cycles until the destination operands of the instruction have been written. |
| Number of µops | 1 per instruction | The number of µops that the instruction is decoded into. |
| ReadAdvance cycles | 7 per instruction | How much the effective latency of each source operand is decreased. |
| PortMap | 23 per instruction | For how many cycles the instruction occupies each execution port or each group of execution ports. |

mca parameters are equally important for achieving a good accuracy on the BHive set, and it might be sufficient to focus on the commonly occurring, simple instructions.

To test this hypothesis, we consider one of the simplest sets of parameter values that is possible: We assume that all instructions are decoded into one µop, and that they have a latency of one cycle; we set all other instruction-specific parameters to 0. We set the two global llvm-mca parameters, the dispatch width and the reorder buffer size, to 4 and 100, respectively. Some insights into why these values were selected are provided in Section III.

Surprisingly, on all four microarchitectures that were considered in the DiffTune paper, this basic set of parameter values leads to predictions that are more accurate than the predictions obtained with both DiffTune's learned parameters and llvm-mca's default parameters; more details can be found in Section V. Note that we use exactly the same parameter values for all four microarchitectures. This shows that it is actually easy to find parameter values that achieve an error that is competitive with DiffTune's learned and llvm-mca's default values on the BHive benchmarks.

Based on these insights, we then propose in Section IV a new learning algorithm for the llvm-mca parameters. Our algorithm first initializes all parameter values with the simple values described previously. Then, it iterates over all instructions and their parameters one-by-one, and checks if a better accuracy can be achieved with a higher parameter value. Unlike DiffTune, our approach does not require differentiability, and can thus be directly used with the original simulator, instead of requiring a differentiable surrogate.

We show in Section V that the parameter values learned with our approach lead to an average error that is between 29% and 47% lower compared to DiffTune's values, even though DiffTune's approach is significantly more complex.

## II. PROBLEM STATEMENT

Given a set $B$ of benchmarks, our goal is to find values for the parameters shown in Table I, such that the average error of llvm-mca's predictions relative to the measured throughput is minimized. The average error, or "mean absolute percentage error" (MAPE) is defined as follows. Let $m(b)$ be the measured throughput of a benchmark $b$, and $p(b)$ llvm-mca's prediction for $b$ for a specific set of parameter values. Then

$$MAPE(B) = \frac{1}{|B|} \cdot \sum_{b \in B} \frac{|m(b) - p(b)|}{m(b)}$$

## III. A BASIC SET OF PARAMETERS

In this section, we provide some further insights regarding our basic set of parameter values that we proposed in the introduction.

We set the dispatch width to 4. This value applies to all microarchitectures that were considered in the DiffTune paper, and it corresponds to the decoding limits that are documented in Intel's and AMD's manuals [5], [10]; note that with the methodology that is used to measure the throughput of the BHive benchmarks [7], the benchmarks are executed in a way in which they are not able to use the µop cache, but have to go through the decode units [4].

We set the reorder buffer size to 100. Renda et al. [12] showed that all values above 70 lead to the same average error on the BHive suite.

For the instruction-specific parameters, we set the latency and the number of µops to 1, and all other parameters to 0. These values typically correspond to a best-case scenario (with a few exceptions due to, e.g., zero-latency moves).

One reason for picking values that correspond to a best-case scenario is that the average error, as defined in Section II, can never be higher than 100% for predictions that are smaller than the measurements, but it can be arbitrarily high for predictions that are too large.

Another reason is that even though the actual values for the parameters might be higher, modern processors implement many optimization that aim to bring the actual performance close to the best case, and thus, smaller values can lead to more accurate predictions than the actual values. An example of such an optimization is "micro fusion". Here, two µops of the same instruction are fused together by the decoders and treated as on µop in the early stages of the pipeline; they are then split again into two µops in the later stages of the pipeline. llvm-mca does not model micro fusion. Therefore, using a µop count that is lower than the actual count can result in a better throughput prediction.

## IV. A SIMPLE LEARNING ALGORITHM

In this section, we describe a fairly straightforward algorithm for learning the parameters of llvm-mca. The corresponding pseudo code is shown in Algorithm 1.

We first initialize all parameters to the values described in Section III (line 1 to line 7). Then, we iterate over all the instructions that occur in the BHive benchmark suite. For each instruction, we select $1,000$ basic blocks from the training

**Algorithm 1:** Simple learning algorithm

---

**1** $par[DispatchWidth] \leftarrow 4$
**2** $par[ReorderBufferSize] \leftarrow 100$
**3** **foreach** *instr in instructions* **do**
**4**    $par[instr][Latency] \leftarrow 1$
**5**    $par[instr][\mu ops] \leftarrow 1$
**6**    **foreach** *p in remainingParameters* **do**
**7**       $par[instr][p] \leftarrow 0$

**8** **foreach** *instr in instructions* **do**
**9**    $blocks \leftarrow 1,000$ BHive blocks that contain *instr*
**10**    $bestError \leftarrow \mathrm{getMAPE}(blocks, par)$
**11**    **foreach** *p in parameters* **do**
**12**       **while** $par[instr][p] < 10$ **do**
**13**          $par[instr][p]$++
**14**          $e \leftarrow \mathrm{getMAPE}(blocks, par)$
**15**          **if** $e < bestError$ **then**
**16**             $bestError \leftarrow e$
**17**          **else**
**18**             $par[instr][p]$--
**19**             break

**20** **return** *par*

---

set that contain this instruction; if there are fewer than $1,000$ such blocks, we select all of them. We then iterate over all the instruction-specific parameters in the order in which they are shown in Table I. We keep increasing the value of each parameter until it reaches 10, or until the average error of the throughput predictions for the $1,000$ basic blocks becomes worse than with the previous value, whichever happens first. Finally, we return the current set of parameter values.

Note that our approach attempts to learn all 23 PortMap parameters (see Table I). DiffTune sets the 13 PortMap parameters that correspond to port groups to zero; port groups are used to model the execution of instructions that have µops that can use more than one port. According to Renda et al., DiffTune sets these parameters to zero because "the simulation of port group parameters in the PortMap does not correspond to standard definitions of port groups", though it is not clear why that would be relevant.

## V. EVALUATION

### A. Methodology

We use the same methodology that was used in the DiffTune paper, which we summarize in the following.

We use the BHive dataset, which was proposed by Chen et al. [7]. This dataset contains $287,639$ basic blocks that were extracted from applications from different domains, such as numerical computation, databases, compilers, machine learning, and cryptography. We use the reference measurements that are published on GitHub[1]. 80% of the blocks of the dataset

---

[1]https://github.com/ithemal/bhive/tree/5878a18/benchmark/throughput

are used as a training set, 10% as a validation set, and the remaining 10% as the test set; we use the same training, validation and test sets as in the DiffTune paper.

The results are evaluated according to the following metrics:

- The mean absolute percentage error (MAPE) of the predictions relative to the measurements, as defined in Section II.
- Kendall's tau coefficient [11], which is a measure for how well the pairwise ordering is preserved.

### B. Results

Table II shows the average error and Kendall's tau coefficient on the test set for the same four microarchitectures that were considered in the DiffTune paper. The rows labeled with "Simple" contain the results for the simple parameter set proposed in Section III; note that we use exactly the same parameter values here for all four microarchitectures. The results for the learning approach described in Section IV can be found in the rows labeled with "Learned". The results for DiffTune were obtained with the learned parameters provided at[2]. Note that the values are slightly different than the values reported in the DiffTune paper. This is because the values in the DiffTune paper are averages over three different runs of DiffTune; unfortunately, according to the authors, the learned parameters of the other two runs have been overwritten and are thus not available any more[2]. We also provide data for the default parameters of llvm-mca (version 8.0.1).

Table III shows the average error on the Haswell microarchitecture of the different tools for the applications, from which the BHive blocks were extracted, and for different categories of basic blocks, such as basic blocks that contain mostly vector instructions (Vec), or basic blocks that contain mostly loads (Ld) or stores (St).

*1) Comparison with DiffTune:* Both of our approaches perform better than DiffTune on all four microarchitectures. For the simple parameter set from Section III, the average error is between 2% and 15% lower. The learning technique from Section IV leads to an average error that is between 29% and 47% lower.

Kendall's tau coefficient for the simple parameter set is slightly better than DiffTune's; for the learning approach, it outperforms DiffTune significantly.

Table III shows that both of our approaches perform better than DiffTune for all source applications and for all benchmark categories.

*2) Comparison with llvm-mca:* Our two approaches lead to a lower average error than llvm-mca's default parameters on all four microarchitectures; DiffTune's parameters lead to a lower error on three of the microarchitectures.

Kendall's tau coefficient of the learning approach described in Section IV is higher than with the default parameters, whereas it is lower for the simple parameter set and for DiffTune.

---

[2]https://github.com/ithemal/DiffTune/issues/1

However, we think that the comparison of llvm-mca's default parameters with DiffTune's and with our parameters should be treated with some caution for the following reasons.

- As pointed out in [4], the reference measurements for the BHive benchmarks contain a significant number of measurement errors. It is therefore possible that the lower error of DiffTune and of our learning approach is partly due to overfitting to these measurement errors.
- As also pointed out in [4], the measurement methodology that was used to obtain the reference measurements is not based on exactly the same throughput definition that llvm-mca uses, and there are a number of benchmarks that have an input-dependent throughput. It is therefore possible that with a different measurement methodology, the accuracy of llvm-mca relative to these measurements would be higher.
- It is not clear whether the results generalize to datasets other than the BHive benchmarks. Certain benchmark types appear to be overrepresented in the BHive suite; for example, there are $12,919$ benchmarks that only consist of a single `cmp` instruction (with different registers, immediates, and memory offsets). On the other hand, benchmarks with vector instructions, which are common in performance-critical code, appear to be underrepresented.
- The goal of llvm-mca is not only to predict the throughput, but to also provide other insights into how the code is executed, such as which execution ports are used by which instructions. On the other hand, the learning approaches only focus on predicting the throughput accurately; the learned parameters may not correspond to the actual parameters used by the hardware. For benchmarks which are bottlenecked by some component that is not modeled by llvm-mca, such as the instruction decoders, the learning approaches might lead to a higher parameter value for some other parameter in such cases that might lead to a better timing prediction, but doesn't provide any actual insight into how the code is executed.

*C. Execution Time*

The learning algorithm from Section IV requires around 14 hours per microarchitecture on an Intel Core i5-12600K.

## VI. CONCLUSIONS AND FUTURE WORK

Neural network-based machine learning techniques are currently a hot research topic. They have proven to be suitable for many different applications. However, it is not clear whether learning llvm-mca parameters is actually one of them — our results show that state-of-the-art neural network-based techniques are not able to beat a trivial set of parameters values, and are outperformed significantly by a simple, straightforward learning algorithm.

It is not known what the optimal solution to the problem of finding llvm-mca parameter values is; an average error close to 0 is probably not achievable just by modifying the instruction-specific parameter values, as there are other performance-relevant aspects of the microarchitecture that

TABLE II: Comparison of different approaches

| $\mu$Arch | Predictor | MAPE | Kendall |
|---|---|---|---|
| Ivy Bridge | llvm-mca-8 | 32.05% | 0.7872 |
| | DiffTune | 25.92% | 0.7244 |
| | Simple | 24.01% | 0.7366 |
| | Learned | 14.35% | 0.8260 |
| Haswell | llvm-mca-8 | 24.77% | 0.7808 |
| | DiffTune | 25.22% | 0.7333 |
| | Simple | 22.51% | 0.7470 |
| | Learned | 13.45% | 0.8331 |
| Skylake | llvm-mca-8 | 26.51% | 0.7721 |
| | DiffTune | 24.73% | 0.7367 |
| | Simple | 21.07% | 0.7477 |
| | Learned | 13.34% | 0.8359 |
| Zen 2 | llvm-mca-8 | 34.90% | 0.7940 |
| | DiffTune | 25.65% | 0.6812 |
| | Simple | 25.06% | 0.7197 |
| | Learned | 18.21% | 0.8196 |

TABLE III: Average error on Haswell for different applications and categories

| Block type | # Blocks | llvm-mca | DiffTune | Simple | Learned |
|---|---|---|---|---|---|
| OpenBLAS | 1478 | 28.77% | 38.01% | 28.47% | 15.44% |
| Redis | 839 | 41.16% | 26.77% | 23.93% | 14.79% |
| SQLite | 764 | 32.83% | 26.29% | 25.24% | 16.09% |
| GZip | 182 | 40.63% | 32.97% | 25.78% | 17.80% |
| TensorFlow | 6399 | 33.47% | 24.12% | 23.36% | 14.89% |
| Clang/LLVM | 18781 | 22.00% | 24.55% | 21.97% | 12.97% |
| Eigen | 387 | 44.35% | 28.03% | 25.16% | 17.19% |
| Embree | 1067 | 34.07% | 27.53% | 22.54% | 13.41% |
| FFmpeg | 1516 | 30.91% | 26.92% | 23.18% | 14.81% |
| OpenSSL | 582 | 36.99% | 27.07% | 23.83% | 15.09% |
| Scalar | 7952 | 17.24% | 23.22% | 18.14% | 12.44% |
| Vec | 104 | 35.28% | 63.14% | 44.15% | 21.36% |
| Scalar/Vec | 614 | 53.56% | 44.15% | 43.72% | 20.58% |
| Ld | 10850 | 27.22% | 28.74% | 27.12% | 14.44% |
| St | 4490 | 24.70% | 11.98% | 9.13% | 6.74% |
| Ld/St | 4754 | 27.91% | 29.77% | 28.74% | 18.13% |

would need to be modeled in order to achieve such an accuracy. Due to the simplicity of our learning algorithm, it is unlikely that it is able to find a solution that is close to the actual optimum. Thus, there is likely a lot of potential for further improvements that could be explored in future work; our results can serve as a baseline for evaluating such future work. Whether neural network-based techniques can be made competitive, or whether other techniques provide a more promising way forward, will remain to be seen.

## ARTIFACTS

An implementation of our learning algorithm, as well as the parameter sets that we used for the evaluation, are available at https://github.com/andreas-abel/DiffTune-Revisited.

## ACKNOWLEDGMENTS

This project has received funding from the European Research Council under the European Union's Horizon 2020 research and innovation programme (grant agreement No. 101020415).

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
