# OpenReview forum: "DiffTune Revisited: A Simple Baseline for Evaluating Learned llvm-mca Parameters"
_iscaconf.org/ISCA/2022/Workshop/MLArchSys — MLArchSys 2022_

### Official Review · Reviewer_QELf · 2022-05-16
**Simpler tuning of llvm-mca parameters**

**Rating:** 6
**Confidence:** 4

**Review:**

The authors show that tuning >10,000 parameters to best predict basic block timing for x86 systems have a concise and simple assignment that is more accurate than the microarchitecture-specific hand tuned parameters, as well as the results obtained by DiffTune (published at MICRO in 2020.)

This paper proposes a simple search technique to refine this concise assignment to further improve the accuracy of llvm-mca.

The primary contributions of this paper are:
* The default llvm-mca parameter settings are a very poor baseline for any comparison.
* DiffTune isn't doing a great job improving the results. But the reason why is not explained.

I do not think the simple search technique that the authors describe is really a contribution. Thus, this paper is primarily a refutation of the DiffTune paper.

I'm not exactly sure how to review a refutation article. I do believe such an article should have results that are as easily reproducible as possible, and that reproducibility should be a requirement for publication. (It is unclear to me as a reviewer whether there are artifacts submitted with this paper.) The authors do not insinuate any bad behavior on the part of the DiffTune authors.

Concerning the title: "DiffTune revisited" implied to me that this might be an extension or improvement to DiffTune. This is inaccurate, but I can see the challenge in not wanting to be too inflammatory. I think perhaps my title above might be more accurate and still non-insinuating.

I was disappointed with the conclusions. Summarizing the section:
* Simulator parameters might not be amenable to optimization using NN-based machine learning. (I think that it is overreach to claim that the entire problem space is not amenable.)
* The optimal solution for this problem is probably unknowable, and the simple search technique is unlikely to find it.

My take-aways are simply:
* It is best practice to do some simple search on a discrete problem like this to determine if your elaborate technique is worth it.
* Don't trust that a published model/simulator is a reasonably accurate one. llvm-mca obviously isn't.

The conclusions I think would be a significant contribution, but is not proven here is:
* It is not a good idea to build a differentiable proxy of a discrete and possible bumpy parameter space. (Unproven)

Issues/Concerns/Questions/Opportunities for improvement:
* There is no description of *how* the authors discovered the simple baseline in Section III. Was this just a hunch? Was this actually there a broader search and this was just a simple way to summarize the weakness of the original baseline?
* I would expect a gradient-trained NN to at least achieve the same results in a continuous problem domain. Is the problem with DiffTune that it builds a differentiable model? Is there any way to prove that?

Concerning my rating:
The work (assuming it is replicable) is good. Science moves by refutation of previous results, and this paper does it carefully. I don't find that it adds much beyond the refutation. It doesn't prove anything about how DiffTune fails that allows for any generalization (if there is something there.) It doesn't really talk about best practices. And the technique described to search is not novel. I believe that these results justify a dialog in the workshop, and here on OpenReview (when this is opened), which might lead to more general results.

---

### Official Review · Reviewer_8SQn · 2022-05-19
**Empirical comparison between heuristics and ML-driven methods for parameter tuning  of llvm-mca (the choice of dataset could bias the results)**

**Rating:** 7
**Confidence:** 4

**Review:**

The paper empirically explores the problem of parameter tuning of llvm-mca (a performance analysis tools in LLVM ecosystem). The authors suggest that for this specific problem setting the simulator parameters in an ad-hoc way (surprisingly) outperforms the accuracy of llvm-mca with default parameters, claiming SOTA accuracy is trivial for this particular problem setting (or possibly in other areas). Based on this intuition, the authors propose a simple (grid) search algorithm to further tune (overfit) the microarchitecture-specific parameters in llvm-mca for a particular dataset (BHive).

The results from this simple proposed search algorithm shows significant improvements in throughput prediction accuracy compared to a ML-Driven method (DiffTune) and even llvm-mca default parameters. Overall, I like the paper and some of the conclusions that the authors made in the paper. However, I think the authors could possibly perform more ablation studies compared to existing methods and provide intuitions and guidelines for other work in this area. I also appreciate the effort that the authors made to highlight the limitations of their study, specifically, talking about the impact of dataset (which I found intriguing).

What I would like to see in the future versions of this paper is to include intuitions/suggestions about how the authors came up with this particular search algorithm, did they use domain knowledge (which I guess is likely to be the case) or it was random search? something that machine learning techniques currently lack.

I would also suggest the authors to add some description about the limitation of the dataset and their observations. I am curious to know whether there are particular instructions or basic blocks that works better for an ML-driven approach or the proposed method consistently outperform existing approaches. Also, it would be more interesting if the authors can further extend the dataset, adding more complex basic blocks, and perform further experiments to study the limitations of each techniques.

It is also good to see whether this simple search algorithm works well on some unseen dataset (generalization of the method). I presume that an ML driven method would possible shine under this setting, provide better generalization to unseen instructions/dataset. I suggest the authors to consider this dimension as well in the future versions of this work.

Finally, I think a transparent release of dataset, simulator, parameters, etc. would benefit the broader scientific community. I would also suggest the authors to discuss their findings with the authors of DiffTune paper (if they have not done so already), this would help to push the boundaries of science further and facilitate future research in this domain.

---

### Official Review · Reviewer_Eg9M · 2022-05-21
**Bad-mannered attack paper with somewhat unsubstantiated claims**

**Rating:** 2
**Confidence:** 5

**Review:**

This paper attacks DiffTune, a neural network-based technique to automatically learn the microarchitecture-specific parameters of basic block CPU simulators. DiffTune was applied to the llvm-mca basic block simulator and previously demonstrated that learned simulator hyperparameters produced better results than the human expert-defined hyperparameters when evaluated on the BHive benchmark suite. For clarity, I will refer to this paper as Revisited, and the original DiffTune paper as DiffTune.

This paper presents a seemingly-trivial solution to setting better llvm-mca hyperparameters by statically setting Dispatch Width to 4, ROB size to 100, and the latency and uops of each instruction to 1. Then, the authors present a simple heuristic for setting all PortMap parameters for each instruction. The authors show that this simple heuristic outperforms the learned technique in DiffTune.

There are two classes of key questions about this paper: firstly, regarding the clarity of the Revisited paper and whether the claims in the paper are accurate. Secondly, whether this style and tone of the Revisited paper is considered acceptable decorum in our field.

**Technical questions/concerns about this paper:**

1. The simple algorithm presented in Revisited learns all 23 PortMap parameters, whereas DiffTune only learns 13 PortMap parameters and sets the remaining to zero. A key question is whether the claimed accuracy improvements are due to the improved PortMap parameter coverage. This paper should have evaluated two variants: one as presented in the Revisited paper (learning all 23 parameters), and another that only learns the same 13 PortMap parameters used in DiffTune (setting the remaining to zero, similar to DiffTune).

2. This is simply a clarity question: it's not entirely clear what the difference is between the authors' presented Simple and Learned algorithms. The section on Simple does not mention PortMap settings. What are the PortMap settings for Simple? Is the difference between Simple and Learned just how it deals with PortMap?

3. The paper presents substantially worse accuracy results for DiffTune relative to what the DiffTune authors presented in their paper (see Table III in this paper, compared to Table V in the main DiffTune paper). It's unclear why this paper was unable to replicate the DiffTune results. In this scenario, the authors should have tried much harder to compare against DiffTune in the best possible light (i.e. using their results from Table V, unless the Revisited authors present substantial evidence proving that the original DiffTune results are not real).

4. It doesn't seem to me that the Revisited paper's broad claims that "that state-of-the-art neural network-based techniques are not able to beat a trivial set of parameters values, and are outperformed significantly by a simple, straightforward learning algorithm" is actually substantiated. Instead, the Revisited authors have only shown that this may be true specifically in the context of llvm-mca running the BHive suite evaluated on a bunch of relatively similar 4-way OOO x86 CPUs. Here are some potential reasons why:
	1. It's likely the case that although DiffTune works well, it was not evaluated on a diverse enough set of x86 CPUs (Ivy Bridge, Haswell, Skylake, Zen 2). These CPU microarchitectures are all relatively similar, e.g. they all have 4-way decode with a standard OOO back-end. Suppose DiffTune and Revisited were re-evaluated on Pentium 4, or Intel Atom Bonnell (an in-order CPU design). I would expect DiffTune to be able to adapt to these substantially different microarchitectural designs, whereas the Revisited static settings would not.
	2. It's likely the case that llvm-mca is a fairly inaccurate CPU microarchitectural performance simulator (relative to Gem5, or other more-detailed simulators). As the DiffTune paper has already shown, llvm-mca is quite insensitive to obviously-important parameters like ROB length and Dispatch Width. The reason that statically setting values might work is due to this insensitivity, which shouldn't exist in more-accurate simulators besides llvm-mca.
	3. It's likely the case that BHive is not a reasonable benchmark suite. As the Revisited authors have already stated, there are 12,919 benchmarks that only consist of a single cmp instruction. It's unclear if the conclusions would hold true on a more representative benchmark suite.

**Style and Tone concerns about this paper**

Aside from the technical issues, one of my major concerns with this paper is its style and tone violating academic decorum. Everyone in academia knows that no paper is perfect; all papers have some problems. Does that mean that the ideas and insights presented in these papers are all therefore invalid? Should we all change jobs from doing research and progressing our field to focus on writing attack papers against others and defending ourselves from other attack papers? Academic research should be collaborative, not toxic and harmful. We should build upon the foundations of others, not destroy others for perceived faults.

That being said, people should also be held responsible for the quality of their work, including cases in which someone has made an overstated claim or created an over-engineered solution when it had a more-trivial solution. Unless in egregious cases in which someone actually fabricated their results or some other serious intentional flaw, a better solution (in my opinion) is to focus the paper more on your own work, while still comparing against prior work. For example, instead of titling the paper "DiffTune Revisited", it could be "A Simple Approach for Estimating CPU Simulator Parameters". Obviously, the entire paper will need to be rewritten - this is not merely a simple title change.

**Conclusion**

For both the technical and non-technical reasons presented as above, I unfortunately must Strongly Reject this paper.

---

### Decision · Program_Chairs · 2022-05-30

**Decision:**

Accept

**Comment:**

The reviews for this paper are mixed. While two reviewers found this work interesting and worthy of discussion, one reviewer suggested changes to the tone and content of the paper. The concerns regarding tone were echoed in further discussions. Therefore, although we believe there is merit to discussing the findings at the workshop, we require the authors to revise their submission as the reviews suggest prior to presentation.